# Association between Personal Social Capital and Loneliness among Widowed Older People

**DOI:** 10.3390/ijerph17165799

**Published:** 2020-08-11

**Authors:** Dongdong Jiang, Yitan Hou, Jinwei Hao, Jiayi Zhou, Junfeng Jiang, Quan Wang

**Affiliations:** 1School of Health Sciences, Wuhan University, Wuhan 430071, China; dongdjiang@whu.edu.cn (D.J.); houyitan@whu.edu.cn (Y.H.); hjnwei@whu.edu.cn (J.H.); 2017302180144@whu.edu.cn (J.Z.); jiang0111@whu.edu.cn (J.J.); 2Global Health Institute, Wuhan University, Wuhan 430072, China

**Keywords:** Chinese widowed older people, loneliness, bonding social capital, bridging social capital

## Abstract

To explore the association between the personal social capital and loneliness among the widowed older adults in China. Data from 1497 widowed older adults were extracted from China’s Health-Related Quality of Life Survey for Older Adults 2018. The Chinese version of the Personal Social Capital Scale (PSCS-16) was used to evaluate the participants’ status of bonding and bridging social capital (BOC and BRC). Loneliness was assessed by the short-form UCLA Loneliness Scale (ULS-8). Multiple linear regression models were established to examine the relationship between social capital and loneliness. The BOC and BRC of rural widowed older people were significantly lower than those of widowed older people in urban areas, while loneliness of rural widowed older people was higher than that of widowed older people in urban areas. The result of the final model showed that loneliness of rural participants was significantly associated with both BOC (B = 0.141, *p* = 0.001) and BRC (B = −0.116, *p* = 0.003). The loneliness of the urban widowed sample had no association with both BOC and BRC (*p* > 0.05). These findings suggested that more social support and compassionate care should be provided to enrich the personal social capital and thus to reduce loneliness of widowed older adults, especially those in rural areas.

## 1. Introduction

Loneliness is one of the worldwide public health issues among older people [1]. Related studies were mainly conducted in developed countries. A nationally representative study reported that 43% of older people in the United States felt lonely [2]; among the 25 European nations, loneliness of older adults was estimated to range from 19.6% to 34.0% [3]. Loneliness can cause a host of damages to both physical and mental health of older people, which may lead to a series of physiological effects and accelerate the ageing process [4]. Evidence has shown that loneliness may increase the chances of coronary disease, stroke [5], dementia [6] and depression [7] for older people. The consequences of chronic loneliness may further lead to the increase of the financial burden and psychological pressure for both older adults and their families. Therefore, it is necessary to understand the factors influencing loneliness among older people. Previous studies have proven that demographic information (e.g., age [3], gender, marital status [8] and living areas [9]), activities of daily living [10] and health behaviors [11] can affect loneliness. In addition, social resources, including social capital, are also related to the loneliness [12].

Social capital was first defined as a mixture of potential or actual resources that is built by a stable network and based on familiar or mutually understanding relationships [13], and it was defined as consisting of collective and individual levels [14]. Many studies have proved that social capital plays a role in social support and social networks at the individual level and in social norms and social trust at the collective level [15,16]. With the consensus of researchers who study social capital, bonding social capital (BOC) and bridging social capital (BRC) have become two novel dimensions of social capital research as the research in this field has progressed [17,18]. BOC refers to trust and cooperation between similar members in certain social demographic factors, with an introvertedly homogeneous network and enhanced exclusivity [15], while BRC refers to connections between residents of communities with different statuses and power [19]. Studies have also identified that social capital and health are entwined and inextricably linked with each other [16,20], and their connection among older people has aroused the interest of many researchers [21,22]. Researchers have reported that social capital was related to the improvement of self-rated health [23,24] and the incidence of some diseases like HIV and heart disease [25]. Besides physical health, social capital affects mental health, such as depression [26] and anxiety [27]. 

However, little research has focused on the relationship between social capital and loneliness, especially among older adults. Fredrica et al. [28] found that, in Western Finland, the association between social capital and loneliness varied across age groups, and low social capital was a risk factor for loneliness. Another study conducted by Fredrica et al. [12] showed that loneliness was highly affected by social capital among the very old (over 85 years old) in Finland and Sweden. Coll-Planas et al. [29] suggested that social capital could reduce loneliness among older people in Spain. Furthermore, research on the convoy model of social relations in older people has proved that older people focus more on emotional intimacy and tend to keep in touch with intimates, such as their spouse and son/daughter [30,31]. Under the premise that a small amount of evidence has shown that the social capital of older people in general was associated with loneliness, we considered how widowed older people lack spousal intimacy, which may increase their loneliness and need for other relationships compared with the general older population, that is, they may require more support from other social capital. Therefore, we assumed that the association between social capital and loneliness of the widowed older people may be related or even strongly related. 

In China, limited research exists on the widowed older people’s social capital, especially the relationship between personal social capital and loneliness. As the world’s most populous and developing country, the population ageing process in China is much faster than in many countries worldwide [32], which means the number of widowed older people in China may be relatively higher than in other countries. According to the data of the sixth national census in China in 2010, the total number of widowed individuals aged over 60 years was 47.5 million, accounting for about 26.89% of older people in China [33] and indicating that widowed older people are worthy of concern. Previous studies have found that the level of loneliness among widowed older people was higher than older persons in other marital statuses [34]. Evidence also proved that widowed older people may possess low levels of social support from families and society [35,36], which indicates that their social capital may also be inadequate. Therefore, this study aimed to examine the association between social capital (BOC and BRC) and loneliness among the Chinese widowed older people, with regard to the rural and urban areas. 

## 2. Materials and Methods 

### 2.1. Research Design and Participants

The data of this study were derived from China’s Health-Related Quality of Life Survey for Older Adults 2018 (CHRQLS-OA 2018) [37]. The CHRQLS-OA 2018 was a cross-sectional survey conducted during the Chinese New Year in 2018, aiming to understand the status of health-related quality of life among Chinese older people aged 60 years old and over. The survey collected information including sociodemographic characteristics, personal social capital, behaviors and lifestyles, health-related quality of life, mental health, coping strategies, loneliness and suicide intention. There were 5442 valid subjects included in the general dataset using online and offline questionnaires. The offline survey’s effective rate was 85.26%. The target population in the present study was widowed older people. Therefore, 1497 widowed older adults were included in the analysis. 

### 2.2. Measures

#### 2.2.1. Demographic Characteristics

In this study, we included the following demographic characteristics: age, gender, educational level, family per capita annual income and current work situation. We also added health conditions (number of chronic diseases and sleep problems) and health-related behaviors (smoking and drinking) as confounding factors in screening the association between personal social capital and loneliness among the widowed older people. 

#### 2.2.2. Assessment of Personal Social Capital

In our study, personal social capital was evaluated by the Chinese version of the Personal Social Capital Scale (PSCS-16). The PSCS-16 has been proved to be reliable and valid in China [38,39]; it consists of 16 items and aims to measure the two key types of social capital, namely bonding social capital (BOC) and bridging social capital (BRC). There are four components used to assess BOC, namely network size, trustworthiness, resource ownership and reciprocity. Likewise, there are also four components evaluating BRC, namely network size, resource ownership, trustworthiness and reciprocity [17]. Each component contains two questions (Cronbach’s alpha = 0.961; KMO = 0.960). A 5-point Likert scale (1 = all, 2 = most, 3 = some, 4 = a few and 5 = none) is employed adopted for these items, in which the total score of each personal social capital (BOC and BRC) ranges from 8 to 40. A higher total PSCS-16 score indicates a lower personal social capital. 

#### 2.2.3. Assessment of Loneliness

The loneliness of the older people was evaluated by the short-form UCLA Loneliness Scale (ULS-8). Hays and Dimatteo designed this scale, which consists of eight items selected from the revised UCLA Loneliness Scale [40,41]. The ULS-8 employs a 4-point Likert scale for item scoring, and the response scale for questions estimating participants’ rating of loneliness is 1 (never), 2 (seldom), 3 (sometimes) and 4 (always). Participants with a higher total score were considered to have a higher level of loneliness. Among these eight items, there are two items which need to be reverse-coded. Many Chinese scholars have confirmed the reliability and validity of the ULS-8 [42,43,44]. 

### 2.3. Statistical Analysis

The data analysis was divided into three steps. The first step was that of the descriptive statistics. In this step, the frequency of all the variables for the sample by demographic characteristics (age, gender, educational level, family per capita annual income and current work situation), health-related behaviors (smoking and drinking) and health condition (sleep problems and the number of chronic diseases) was calculated. Chi-square tests and t-tests were applied to examine the statistical difference of demographic characteristics and health-related behaviors between rural and urban areas. Second, t-tests were employed to examine the difference in each dimension of personal social capital and loneliness between rural and urban areas. Finally, multiple linear regression models were established to explore the association between personal social capital and loneliness after controlling for variables like demographic characteristics and health-related behaviors. There were four models established in the final step, which involved one crude model and three adjusted models. 

The Statistical Package for the Social Sciences (SPSS) version 22.0 (SPSS Inc., Chicago, IL, USA) was used to conduct the analyses, with a significance level of 0.05.

### 2.4. Ethical Statement

China’s Health-Related Quality of Life Survey for Older Adults 2018, designed and conducted by the Global Health Institution of Wuhan University, was permitted by the Institutional Review Board (IRB), School of Health Sciences and Faculty of Medical Sciences, Wuhan University. The IRB number of this survey is 2019YF2050. 

The related informed consent information of the survey was introduced to the participants before the investigation. Participants in our survey were those who fully agreed to the contents and purpose of this study. Before conducting the survey, all the investigators were trained and asked to tell the participants that the data would be kept confidential and only used for scientific data analysis.

## 3. Results

### 3.1. General Characteristics of the Study Population

Overall, this study consisted of a total of 1497 widowed older adults, among whom 34.24% were males and 65.76% were females. Among these respondents, the mean age was 74.74 (SD = 7.96). Illiteracy accounted for nearly half of the study population, and the general educational level of widowed older people in rural areas was lower than that in the cities. The family per capita annual incomes of less than 15,000 yuan (45.85%) and between 15,000 and 30,000 yuan (28.66%) accounted for the majority of the sample. It was seen that 20.41% of all respondents were still working, with respondents in rural areas working at a greater proportion than their urban counterparts. Most individuals did not smoke (70.09%), drink (62.88%) or have sleep problems (56.23%). Of the 1497 participants, 580 respondents (40.31%) had no chronic disease, 390 respondents (27.10%) had only one chronic disease and 469 respondents (32.59%) had two or more diseases. 

As shown in Table 1, all subjects were divided into two groups: 981 (65.53%) in rural areas and 501 (33.47%) in urban areas. The following characteristics were found to be significantly different across these two groups: gender (χ^2^ = 8.317, *p* = 0.004), educational level (χ^2^ = 213.918, *p* < 0.001), family per capita annual income (χ^2^ = 202.889, *p* < 0.001), current occupation (χ^2^ = 21.227, *p* < 0.001) and sleep problems (χ^2^ = 14.152, *p* < 0.001).

### 3.2. Distribution of Personal Social Capital Level and Loneliness among Participants from Rural Areas and Urban Areas

Table 2 shows the participants’ scores in two dimensions of personal social capital and loneliness. The levels of the two dimensions (BOC and BRC) among rural respondents were both significantly lower than those of the urban respondents (BOC in rural and urban areas, t = 9.996, *p* < 0.001; BRC in rural and urban areas, t = 10.030, *p* < 0.001). The total score of loneliness among the widowed older adults exhibited a significant difference between rural and urban areas (t = 2.130, *p* = 0.033), and rural participants’ loneliness was higher than that of urban ones.

### 3.3. Relationship between Personal Social Capital and Loneliness

To further examine whether personal social capital was related to loneliness among the widowed older people, multiple linear regression models were conducted. The detailed outcomes of four models are listed in Table 3. Overall, after controlling for other covariates (demographic information and health behaviors), loneliness was only associated with lower BOC scores in the widowed older people (B = 0.108, S.E. = 0.033, *p* = 0.001). The loneliness of participants from rural areas was significantly associated with both BOC (B = 0.141, S.E. = 0.041, *p* = 0.001) and BRC (B = −0.116, S.E. = 0.039, *p* = 0.003). The loneliness of the urban widowed sample had no significant association with both BOC and BRC (*p* > 0.05).

## 4. Discussion

This study found that, for the whole sample, only a lower BOC was related to a higher level of loneliness among Chinese widowed older people. BOC is a description of the size of the social network and the trust and help obtained from neighbors and friends [38]. It is understandable that Chinese people highly value relationships, which are also called “guanxi” in China. “Guanxi” is a unique network concept embedded in Chinese culture, which can be described as concentric circles in which close family members like one’s spouse and descendants are in the center and distant kith, colleagues, etc. are on the periphery based on intimacy and trust [45]. Chinese older people put greater emphasis on “guanxi”, especially the relationship of relatives, due to the traditional culture. Therefore, the level of loneliness of widowed older people is likely to be reduced by encouraging their relatives to visit them more. In addition, visitors should also satisfy the material and spiritual needs of the widowed older people as much as possible, for example, by providing adequate financial support, keeping them company and giving enough respect [46]. 

In our study, BOC and BRC among widowed older people had significant differences between urban and rural areas, and both of them were lower in rural areas than in urban areas. This result was inconsistent with previous studies in which the social capital in rural areas should be richer than cities [47,48]. We speculate that this may be caused by the current dual urban–rural structure in China. More and more rural jobs are shifting to the cities, leading to a decrease of rural resources [49]. Besides, the degree and quantity of group activities among rural residents were lower than those of urban citizens because of the economic gap and the widespread existence of virtual social networks in cities [50], which may also be the reason why rural social capital was lower than that of urban areas. Moreover, because of the implementation of the Chinese urbanization promotion strategy, young people who used to live in the countryside now prefer to make a living in the city [51], resulting in a large number of older adults and children left behind in the countryside. In addition to having lower socioeconomic statuses than the urban older people, the rural older people, particularly the widowed, may also not get timely support from their families. According to the theory of the convoy model of social relations, the most important relationship for the rural older people, especially those who have lost their spouse, is the close relationship with their children [30,31]. Therefore, in addition to encouraging their children or other relatives to increase the frequency of visits, it is highly important to improve the quality of the relationship between the widowed older people and their offspring, such as by strengthening parent–child emotional communication, daily life care and financial support to the widowed old people. 

In this study, BOC was related to loneliness solely for the widowed older people in rural areas. This result is consistent with a longitudinal study in South Korea which found that rich social capital could reduce the level of loneliness [52]. In general, people in rural areas pay more attention to “guanxi” than citizens, especially older adults. The “guanxi” in rural population is illustrated as clan or kinship relations. As a part of traditional Chinese culture, the clan can protect kinship’s interests and satisfy their psychological and emotional needs [53]. Although more and more young people move to the cities, causing the power of clan to become tenuous, the clan still plays an important role in rural areas [54], which may have great effects on the rural widowed older people because their connections with offspring are further weakened after the loss of the spouse. While it is true that a spouse can alleviate physical and psychological pain, the loss of the spouse can lead to some unhealthy behaviors and psychological problems, such as drinking, smoking and depression [55,56]. Evidence showed that strong BOC could have a positive impact on health through psychological mechanisms, like reducing pressure and promoting self-efficacy [25]. Therefore, with regard to the widowed older people, despite the loss of their intimate spouse, it is helpful for their children, kin, friends or neighbors to give them more support, for instance, by making regular visits or phone calls and expressing their care for the older people.

Our study showed that there was no link between BRC and loneliness among the whole population widowed older people. Intriguingly, we found that higher BRC was related to the higher loneliness among widowed older people in rural areas but had no association with the loneliness in urban widowed older people. BRC represents heterogeneous and external contact, trust or reciprocity of individuals from formal or informal social groups [57]. Generally speaking, participants with a richer BRC usually have a lower level of loneliness [52]. It is understandable that older people living in the countryside seldom have chances to contact friends or relatives living outside, while it is easier for urban older people to visit unfamiliar neighbors and contact friends. However, the present study showed a different result. We speculated that it is probably related to the dark side of social capital, which was claimed to be “a source of strain, leading to conflicts, envy and disappointment, thus affecting health negatively” [25,57,58,59]. Although there is an increasing number of studies that demonstrate the great benefits of social capital on health, the effects of social capital on health outcomes are not always positive. Previous studies have indicated that social capital may cause unhealthy behaviors and worsen mental health due to the dark side of social capital [59,60]. Our results indicated that rural widowed older people with higher BRC seek more help from social organizations but also suggested they get less support from close relationships, which may cause them to feel more lonely. In addition, the isolation and limited access to public services in rural areas [61] may lead to a lower possibility for rural widowed older people to communicate with others outside the village. However, as more and more older people have started to use smartphones and the Internet [62], their channels to communicate with the outside world have become more diverse. The use of social communication tools allows them to contact relatives and make friends, which may lead to the increase of BRC. However, the positive effect of communication technologies in social connection and social support is short-term [63]. Therefore, the long-term use of online technologies might not reduce the loneliness of older adults. The urban widowed older people have more potential social relationships; for example, many urban residents migrate from other areas, and thus they can build their relations in the new city and in the original areas [47]. Therefore, urban widowed older adults may have less chance to feel loneliness. It is recommended for rural widowed older people to participate in social activities and use social communication tools appropriately. Local village committees can establish psychological counselling centers for rural widowed older people or send more volunteers to visit them and provide health information. In addition, considering that spouse and offspring can help alleviate the loneliness, local village committees can also host blind dates for the widowed older adults and encourage the younger generation to offer emotional support to their widowed parents.

## 5. Limitations

There are still some limitations that should be noted in our study. Firstly, the use of cross-sectional data reflects the association between social capital and loneliness among Chinese widowed older people in a limited period and does not allow for the identification of causality. Secondly, this investigation depends on the self-report questionnaire, which may lead to certain recall bias. Thirdly, the types and measurements of social capital are still controversial. Each instrument may have limitations and cannot cover all fields of social capital. Finally, our survey did not involve the length of time that the widowed older people had been widowed, which may also have an impact on their health outcomes. According to the existing literature [56,64], the cause-specific mortality and mental health of widowed old people will change over time. Thus, these limitations should be taken into consideration in future studies.

## 6. Conclusions

This study mainly revealed that (1) higher BOC was related to lower loneliness for widowed older people in China, (2) higher BOC had a relationship with lower loneliness among rural widowed older people and (3) higher BRC of widowed older people was associated with higher loneliness in rural areas. Policymakers may formulate strategies based on the relationship between BOC, BRC and loneliness to help widowed older people to attain more social support and compassionate care and further promote their health status improvement. For example, to increase the BOC, it is better to encourage the younger generation and kin to give more support to widowed older people in rural areas. Local village committees should carry out appropriate social activities for the widowed older people, and local primary medical institutions can offer psychological counselling to alleviate the loneliness of widowed older people.

## Figures and Tables

**Table 1 ijerph-17-05799-t001:** Characteristics of the study population.

Variable	Total (*n* = 1497)	Rural Areas (*n* = 981)	Urban Areas (*n* = 501)	χ^2^/*t*	*p*-Value
Categorical Variables					
Gender (%)				8.317	0.004
Male	504 (34.24)	358 (36.79)	146 (29.26)		
Female	968 (65.76)	615 (63.21)	353 (70.74)		
Education level (%)				213.918	<0.001
Illiteracy	692 (46.98)	560 (57,38)	132 (26.56)		
No formal education	121 (8.21)	71 (7.27)	50 (10.06)		
Elementary school	332 (22.54)	220 (22.54)	112 (22.54)		
Middle school	161 (10.93)	84 8.61)	77 (15.49)		
High school or above	167 (11.34)	41 (4.20)	126 (25.35)		
Family per capita annual income (RMB) (%)				202.889	<0.001
≤15,000	664 (45.85)	531 (55.31)	133 (27.25)		
15,000–30,000	415 (28.66)	292 (30.42)	123 (25.20)		
30,000–45,000	173 (11.95)	73 (7.60)	100 (20.49)		
>45,000	196 (13.54)	64 (6.67)	132 (27.05)		
Current occupation				21.227	<0.001
Without occupation	1170 (79.59)	744 (76.15)	426 (86.41)		
With occupation	300 (20.41)	233 (23.85)	67 (13.59)		
Smoking				3.653	0.061
No	963 (70.09)	626 (68.42)	337 (73.42)		
Yes	411 (29.91)	289 (31.58)	122 (26.58)		
Drinking				1.907	0.184
No	830 (62.88)	540 (61.57)	290 (65.46)		
Yes	490 (37.12)	337 (38.43)	153 (34.54)		
Sleep problems				14.152	<0.001
No	781 (56.23)	554 (59.76)	227 (49.13)		
Yes	608 (43.77)	373 (40.24)	235 (50.87)		
Number of chronic diseases				3.534	0.171
0	580 (40.31)	380 (39.62)	200 (41.67)		
1	390 (27.10)	251 (26.17)	139 (28.96)		
≥2	469 (32.59)	328 (34.20)	141 (29.38)		
Metric variables					
Age	74.74 ± 7.96	74.52 ± 7.72	75.20 ± 8.42	−1.536	0.125

NOTE: Sample sizes of the demographic characteristic variables may not sum to *n* = 1497 due to missing values. According to variables sorted by Table 1, the missing data values are 25, 24, 49, 27, 123, 177, 108, 58, 18 and 89, respectively.

**Table 2 ijerph-17-05799-t002:** The distribution of bonding/bridging social capital level and loneliness among participants from rural and urban areas.

Variable	Total (*n* = 1497)	Rural Area (*n* = 981)	Urban Area (*n* = 501)	*t*	*p*-Value
Personal social capital					
Bonding capital	27.50 ± 7.05	28.75 ± 6.99	25.08 ± 6.52	9.996	<0.001
Bridging capital	30.70 ± 7.47	32.04 ±7.26	28.06 ± 7.16	10.030	<0.001
Loneliness					
Mean ± SD	17.30 ± 5.09	17.51 ± 5.08	16.91 ± 5.09	2.130	0.033

**Table 3 ijerph-17-05799-t003:** Multiple linear regression model testing the association between personal social capital and loneliness.

Classification		Total		Rural Areas		Urban Areas	
Personal Social Capital		BOC	BRC	BOC	BRC	BOC	BRC
Model 1	B	0.128	−0.026	0.153	−0.112	0.081	0.096
	S.E	0.033	0.031	0.041	0.040	0.058	0.052
	*p*	<0.001	0.403	<0.001	0.005	0.162	0.062
Model 2	B	0.110	−0.043	0.138	−0.111	0.077	0.059
	S.E	0.031	0.028	0.038	0.036	0.053	0.047
	*p*	<0.001	0.127	<0.001	0.002	0.147	0.208
Model 3	B	0.112	−0.026	0.124	−0.095	0.082	0.085
	S.E	0.031	0.03	0.039	0.038	0.054	0.049
	*p*	<0.001	0.379	0.001	0.012	0.127	0.084
Model 4	B	0.108	−0.035	0.141	−0.116	0.077	0.091
	S.E	0.033	0.030	0.041	0.039	0.058	0.050
	*p*	0.001	0.251	0.001	0.003	0.185	0.072

Note: B, coefficient; S.E, standard Error. Model 1: crude model of BOC and BRC (total: R = 0.111; rural area: R = 0.110; urban area: R = 0.164; 0 < *p* < 0.05). Model 2: adjusted for age, sex, educational level, family per capita annual income (RMB) and current occupation (total: R = 0.183; rural area: R = 0.192; urban area: R = 0.254; 0 < *p* < 0.05). Model 3: adjusted for smoking, drinking, sleep problems and number of chronic diseases (total: R = 0.171; rural area: R = 0.153; urban area: R = 0.246; 0 < *p* < 0.05). Model 4: adjusted for age, sex, educational level, family per capita annual income (RMB), current occupation, smoking, drinking, sleep problems and number of chronic diseases (total: R = 0.230; rural area: R = 0.236; urban area: R = 0.354; 0 < *p* < 0.05).

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
