# Peer review of "Association between Personal Social Capital and Loneliness among Widowed Older People"

_ijerph, 2020, doi:10.3390/ijerph17165799_

Round 1
Reviewer 1 Report
Thanks very much for this manuscript on the loneliness of widowed older people. Please kindly find my recommendations below:
- In page 2 line 62-66, the theoretical justification to investigate the loneliness of widowed older people can be further strengthened. I suggest the authors to employ Convoy Model of Social Relations to explain the importance of lack of intimacy and diminished social support in relation to loneliness.
- Page 3 line 89 to 90, given 93 samples lack information about loneliness, how did the analysis treat them?
- For the multiple linerar regression, please show the change of R square and whether the change is significant.
- P. 6 line 196-198, page 7 line 204-206, the discussion mainly focus on quantity of social network. However, the concept of intimacy and trust in BOC is more appropriate to be addressed by the quality of relationship. Therefore, I am not convinced by the suggestion of increasing the frequency of visit or activities, especially it is impossible for Chinese when the younger generation migrate to urban areas and can only return to rural home once a year during the spring time. Therefore, it is more important to think about how to enhance the quality of relationships for rural widowed older people. Activities have to be meaningful for them as well. Is there any more important implicatoins in the planning of rural area and urban area to strengthen the quality of relationship? This will make this manuscript more meaningful.
- For a manuscript about gerontology, we employ the term older people. Elderly is discriminative for gerontologist.
Author Response
Dear reviewer
Thank you for this supportive comment. We have made a revision according to your suggestions.
Response to Reviewer 1 Comments
Reviewer 1
Point 1: In page 2 line 62-66, the theoretical justification to investigate the loneliness of widowed older people can be further strengthened. I suggest the authors to employ Convoy Model of Social Relations to explain the importance of lack of intimacy and diminished social support in relation to loneliness.
Response 1: We thank and agree with the reviewer for this comment. We have added the context of explaining importance of lack of intimacy and diminished social support in relation to loneliness by using the Convoy Model of Social Relations.
Point 2: Page 3 line 89 to 90, given 93 samples lack information about loneliness, how did the analysis treat them?
Response 2: We deleted this sentence, because the author forgot to delete it during the final review. We have previously averaged the data under the advice of Assoc. Prof. Dr. Liu from Fujian Medical University (we thank for his contribution in the acknowledgement).
Point 3: For the multiple linerar regression, please show the change of R square and whether the change is significant.
Response 3: We have made the changes accordingly.
Point 4: P. 6 line 196-198, page 7 line 204-206, the discussion mainly focus on quantity of social network. However, the concept of intimacy and trust in BOC is more appropriate to be addressed by the quality of relationship. Therefore, I am not convinced by the suggestion of increasing the frequency of visit or activities, especially it is impossible for Chinese when the younger generation migrate to urban areas and can only return to rural home once a year during the spring time. Therefore, it is more important to think about how to enhance the quality of relationships for rural widowed older people. Activities have to be meaningful for them as well. Is there any more important implicatoins in the planning of rural area and urban area to strengthen the quality of relationship? This will make this manuscript more meaningful.
Response 4: We thank the reviewer for this excellent suggestion and have added new content accordingly.
Point 5: For a manuscript about gerontology, we employ the term older people. Elderly is discriminative for gerontologist.
Response 5: We have re-edit the term accordingly through the text.
Reviewer 2 Report
The subject is of great interest to the area. The study was well conducted. Nevertheless, to improve the quality of the paper some issues should be addressed:
- Review the writing;
- Exclude the etc...and so on. Include all the variables that entered the study
- Because this study aimed to examine the association between social capital (BOC and BRC) and loneliness among the Chinese widowed elderly, with regard to the rural-urban areas, missing data in one of those variables is unacceptable. What is the point in maintaining them in the study? They will influence the results in the univariate analysis, but they won't be part of the multivariate analysis.
- The authors should justify the use of the variable "number of children" as a covariate or exclude it. This information is part of the "bonding social capital", which is one of the outcomes being assessed.
- The associations between loneliness and bonding social capital and bridging social capital should be clearly described in the abstract, results, discussion, and conclusion sections.
- The unexpected association between loneliness and higher level of bridging social capital in the urban areas should be better explored in the discussion section.
- Other small issues are pointed out in the manuscript.

Author Response
Dear reviewer:
Thank you for these supportive and valuable review comments. We have made a revision according to your suggestions.
Response to Reviewer 2 Comments
Reviewer 2
The subject is of great interest to the area. The study was well conducted. Nevertheless, to improve the quality of the paper some issues should be addressed:
Point 1: Review the writing
Response 1: We processed the language again and invited native speaker to review it.
Point 2: Exclude the etc...and so on. Include all the variables that entered the study. Because this study aimed to examine the association between social capital (BOC and BRC) and loneliness among the Chinese widowed elderly, with regard to the rural-urban areas, missing data in one of those variables is unacceptable. What is the point in maintaining them in the study? They will influence the results in the univariate analysis, but they won't be part of the multivariate analysis. The authors should justify the use of the variable "number of children" as a covariate or exclude it. This information is part of the "bonding social capital", which is one of the outcomes being assessed.
Response 2: We deleted this sentence, because the author forgot to delete it during the final review. We have previously averaged the data under the advice of Assoc. Prof. Dr. Liu from Fujian Medical University (we thank for his contribution in the acknowledgement). We also deleted the “number of children” and redone the results.
Point 3: The associations between loneliness and bonding social capital and bridging social capital should be clearly described in the abstract, results, discussion, and conclusion sections.
The unexpected association between loneliness and higher level of bridging social capital in the urban areas should be better explored in the discussion section.
Response 3: We have re-organized the related information in the revised manuscript as suggested accordingly.
Point 4: Other small issues are pointed out in the manuscript.
Response 4: The standard of invalid samples is that the respondents’ answer time is far shorter than the normal answer time and there are more than 15% of the answers to questions, reviewers can not clearly understand the respondents’ opinion about the question through the manual interpretation. Therefore “5442 valid samples” does not mean the questionnaire with complete data. Other issues have re-edited in the paper.
Reviewer 3 Report
Thank you for the opportunity to review this manuscript. The following are corrections that need to be made to the script.
- Remove 'etc'...from the following lines 34, 38,39,55, and replace either with the correct citations or remove altogether.
- Remove 'and so on' from line 86. .
- Laura et al on line 61 is an incorrect in text citation, please correct with the right authors name.
- Discussion - this needs to be revised and the word 'old individuals' should be replaced with the word 'elderly'. The sentence beginning with 'Although more young people... (lines 217-221) does not make sense please revise. The sentence beginning with 'According to the definition of BRC... (lines 230-232) again does not make sense, please revise. The sentence beginning with However, old urban widowed.. (lines 243-245) does not make sense, do you mean many urban residents who migrate from other areas have the opportunity to build new relationships? If so what promotes this? And where is the evidence for this claim? Line 251 'old individuals' should be replaced with the term elderly.
- References: no 30, 33, 45,57, need to be referenced correctly.
Author Response
Dear reviewer
Thank you for these supportive comments. We have made a revision according to your suggestions.
Response to Reviewer 3 Comments
Reviewer 3
Point 1: Remove 'etc'...from the following lines 34, 38,39,55, and replace either with the correct citations or remove altogether.
Remove 'and so on' from line 86. .
Line 251 'old individuals' should be replaced with the term elderly.
Response 1: We have made this edit.
Point 2: Laura et al on line 61 is an incorrect in text citation, please correct with the right authors name.
Response 2: We have made it correct.
Point 3: Discussion - this needs to be revised and the word 'old individuals' should be replaced with the word 'elderly'. The sentence beginning with 'Although more young people... (lines 217-221) does not make sense please revise.
Response 3: We have revised the sentence and made it more clear now.
Point 4: The sentence beginning with 'According to the definition of BRC... (lines 230-232) again does not make sense, please revise.
Response 4: We have made this revision.
Point 5: The sentence beginning with However, old urban widowed.. (lines 243-245) does not make sense, do you mean many urban residents who migrate from other areas have the opportunity to build new relationships? If so what promotes this? And where is the evidence for this claim?
Response 5: According to the reference (Sorensen, J. F. L., Rural-Urban Differences in Bonding and Bridging Social Capital. Reg. Stud. 2016, 50, (3), 391-410.), urban residents who migrate from other cities can “besides building relations in their new city, they may carry on having contact with people that they have met and gotten to know in other geographical setting.”, because “city life is indeed more diversified than rural life” and “there may be more ethnicities, a bigger variation in terms of income and an abundance of different leisure-time activities dispersed around the city. These more diverse urban impulses may produce a more outward looking perspective on life, along with a more diversified interest in different aspects of life in general”.
Point 6: References: no 30, 33, 45,57, need to be referenced correctly.
Response 6: We have re-edited these references.
Round 2
Reviewer 2 Report
The manuscript has improved greatly. However, the authors changed the interpretation of the UCLA loneliness scale, considering that higher scores are indicative of lower levels of loneliness. Hence, the abstract, the results (difference in loneliness between rural and urban older people and the direction of the associations in the multivariate analyses), and the discussion section should be reviewed accordingly. Also, the description of the results should be reviewed in the results section, considering that the multivariate analyses were redone. Other small issues are pointed out in the manuscript.

Author Response
Dear reviewer
Thank you for these supportive comments. We have made a new revision according to your suggestions.
Response to Reviewers’ Comments
Point: The manuscript has improved greatly. However, the authors changed the interpretation of the UCLA loneliness scale, considering that higher scores are indicative of lower levels of loneliness. Hence, the abstract, the results (difference in loneliness between rural and urban older people and the direction of the associations in the multivariate analyses), and the discussion section should be reviewed accordingly. Also, the description of the results should be reviewed in the results section, considering that the multivariate analyses were redone. Other small issues are pointed out in the manuscript.
Response:
Dear reviewer:
We very appreciate your valuable and constructive comments on this manuscript. We are also very thankful for your careful review of every sentence, which avoids our mistakes if the paper is published. About your comment of “the authors changed the interpretation of the UCLA loneliness scale, considering that higher scores are indicative of lower levels of loneliness”, we are so sorry for our carelessness. Before submitting the revision, we did a language re-editing. Due to certain misleading communication between the language-reviewer and us, we made the current mistake. We now correct the mistake and update the manuscript according to your suggestions. We, all of the co-authors, re-review the article hoping to avoid the other mistakes.
You can find details what we have re-edited in:
- line 48, we delete a Space in “[13],”;
- line 63, we replace the word “research” with “researches”;
3) line 103-104, we replace the word “was” with “in”;
4) line 127-129, we delete the sentence “Notably, the… ”;
5) 2.2.3. Assessment of loneliness line 133-135, we replace the original sentence with the new sentence “Participants with a higher total score were considered to have a higher level of loneliness.”;
6) 3.3. Relationship between personal social capital and loneliness line 199-204, we have changed the value according to the table;
We have re-discussed the results which were redone due to the removal of the variable “number of children”, you can check in the discussion section, line 269-320
Thank you again for your excellent reviewing work.
Best wishes!